# Peer review of "Clinical Applications and Future Directions of Smartphone Fundus Imaging"

_diagnostics, 2024, doi:10.3390/diagnostics14131395_

Round 1

Reviewer 1 Report

Comments and Suggestions for Authors

The authors provided an overview of smartphone fundus imaging, including clinical applications, advantages, limitations, clinical applications and future directions. 

1. The authors should compare with other related reviews, such as references [10], and illustrate the main differences.

2. The formatting of tables is not easy to read.

3. About the diseases, the authors should further search related papers, such as AMD. The following paper is related, but the authors do not refer to 4.1.3.

Sayadia SB, Elloumi Y, Kachouri R, Akil M, Abdallah AB, Bedoui MH. Automated method for real-time AMD screening of fundus images dedicated for mobile devices. Med Biol Eng Comput. 2022 May;60(5):1449-1479. doi: 10.1007/s11517-022-02546-8. Epub 2022 Mar 18. PMID: 35304672.

Comments on the Quality of English Language

1. The formatting of cited references should be revised.

Author Response

Reviewer #1

The authors provided an overview of smartphone fundus imaging, including clinical applications, advantages, limitations, clinical applications and future directions. 

  1. The authors should compare with other related reviews, such as references [10], and illustrate the main differences.

→ We found four seminal review papers on the relevant reviews, including Reference #10. Here is the list of the review papers, which we cited in our manuscript:

Wintergerst, M.W.M.; Jansen, L.G.; Holz, F.G.; Finger, R.P. Smartphone-Based Fundus Imaging-Where Are We Now? Asia Pac J Ophthalmol (Phila) 2020, 9, 308-314, doi:10.1097/APO.0000000000000303.

Iqbal, U. Smartphone fundus photography: a narrative review. Int J Retina Vitreous 2021, 7, 44, doi:10.1186/s40942-021-00313-9.

Prayogo, M.E.; Zaharo, A.F.; Damayanti, N.N.R.; Widyaputri, F.; Thobari, J.A.; Susanti, V.Y.; Sasongko, M.B. Accuracy of Low-Cost, Smartphone-Based Retinal Photography for Diabetic Retinopathy Screening: A Systematic Review. Clin Ophthalmol 2023, 17, 2459-2470, doi:10.2147/OPTH.S416422.

Hunt, B.; Ruiz, A.; Pogue, B. Smartphone-based imaging systems for medical applications: a critical review. J Biomed Opt 2021, 26, doi:10.1117/1.JBO.26.4.040902.

We carefully reviewed all the review papers and compared them with ours to find the main focus and contents of each review paper and how ours is different from previous ones. Overall, our review goes into much greater depth on the various clinical use cases of smartphone fundus imaging and recent findings for different ophthalmic diseases and retinopathy conditions like diabetic retinopathy and retinopathy of prematurity. It also discusses the potential future applications enabled by advancements in smartphone cameras, artificial intelligence, and teleophthalmology.

We have generated a table outlining the reference number in this manuscript, authors, years, contents, and main focus of the paper. In the table, underlined texts indicate the areas where this review is different from previous ones and thus adds to the literature.

Supplementary Table S1. Summary of content and main focus in key review articles. Underlined texts indicate areas where this review differs from previous ones, adding unique contributions to the literature.

Reference No.

Authors

Year

Contents

Main focus

10

Iqbal

2021

Optics of smartphone retinal imaging, Techniques and devices of smartphone retinal imaging, Safety

Techniques of imaging

3

Wintergerst et al.

2020

Overview of smartphone-based fundus imaging, current applications, and limitations

Current state of smartphone fundus imaging

17

Prayogo et al.

2023

Accuracy of smartphone-based retinal photography for diabetic retinopathy screening

Diabetic retinopathy screening

25

Hunt et al.

2021

Smartphone-based imaging systems for medical use, critical review of various systems

Methods and material of smartphone imaging

Ours

2024

Advantages and limitations, techniques and tips, clinical applications (including conventional and novel indications), and future directions

Clinical applications: including conventional and novel indications

We have added this table as Supplementary Table S1, as we believe that this information may be useful for the readers to understand what contents are included in each review paper and how our review paper is unique and adds to the literature. Furthermore, we have highlighted that clinical (conventional and novel) applications are our main focus, which is discriminating feature of our review from previous ones.

In the Introduction section, we have added the sentences “There have been several efforts to apply this technique to various ophthalmic diseases. However, this has not been outlined over diverse retinal diseases, as techniques and specific indications have been addressed in previous research and review papers. Therefore, this review paper aims to fill this gap by providing a comprehensive analysis of the clinical applications, conventional and novel ones, and future directions of smartphone fundus imaging in retinal diseases, together with an overview of smartphone fundus imaging, including its advantages, limitations, and techniques. From the existing literature and evidence, we aimed to elucidate the potential of smartphone fundus imaging as a valuable tool that can be more widely used in the practice of managing retinal diseases.” (Lines 57-66)

Additionally, we have created the section '4.4. Novel Clinical Applications' to highlight the latest advancements in smartphone fundus imaging, which distinguishes our review from others in the field.

  1. The formatting of tables is not easy to read.

→ We have meticulously reviewed all tables to enhance their readability. In Table 1, we have streamlined the text by eliminating non-essential details while preserving key topics. Table 2 has been refined by removing unnecessary items and succinctly summarizing the main outcome measures. These adjustments have significantly improved the tables' readability.

The tables now appear as follows:

Table 1. Advantages and limitations of smartphone fundus imaging

Advantages

Limitations

Accessibility and portability

Cost-effectiveness

Familiarity and improved patient engagement

Continuous improvements and updates by continuous software and hardware improvements

Limited image quality, resolution, and field of view

Lack of standardization and calibration leading to variability in image quality and diagnostic accuracy

Regulatory issues for medical devices

Medicolegal issues such as patient privacy, data security, risk of unauthorized access, and potential breaches of confidentiality

Table 2. Summary of clinical applications of smartphone fundus imaging for retinal diseases in previous studies

Clinical Application

Study (year)

Main Outcome Measures

Findings

Diabetic Retinopathy (DR)

Ryan et al. (2015)[1]

Diagnostic capability for DR detection

Sensitivity: 50%, specificity: 94% (81% and 94%, respectively, using non-mydriatic fundus camera)

Russo et al. (2015) [2]

Diagnostic capability for clinically significant macular edema

Sensitivity: 81%, specificity: 98%

Toy et al. (2016) [3]

Diagnostic capability for  moderate nonproliferative and worse diabetic retinopathy dectection

Sensitivity: 91%, specificity: 99%  

Wintergerst et al. (2020)[4]

Diagnostic capability for DR detection

Sensitivity/specificity: 0.79/0.99 for any DR and 1.0/1.0 for severe DR, 0.79/1.0 for diabetic maculopathy

Gobbi et al. (2022)[5]

Diagnostic capability for DR detection

Sensitivity: 71%, Specificity: 94%

Retinopathy of Prematurity (ROP)

Goyal et al. (2019)[6]

Image quality of wide-field smartphone imaging

Good image quality in 89.3%

Wintergerst et al. (2019)[7]

Diagnostic capability for ROP and plus disease

Sensitivity: 90%, Specificity: 100% for plus disease

Sensitivity: 88%, Specificity: 93% for ROP

Patel et al. (2019) [8]

Agreement between the gold standard and photographic assessment of presence of plus disease

Substantial agreement (Cohen's kappa = 0.85)

Lin et al. (2022) [9]

Agreement between binocular indirect ophthalmoscopy and smartphone-based imaging on plus disease

Moderate agreement (Cohen's kappa = 0.619)

Age-related macular degeneration

Sayadia et al. (2022) [10]

Diagnostic accuracy

From 94.3% to 100%, depending on the database used.

Hypertensive retinopathy

Sajid et al. (2023) [11]

Diagnostic accuracy

99%

  1. About the diseases, the authors should further search related papers, such as AMD. The following paper is related, but the authors do not refer to 4.1.3.

Sayadia SB, Elloumi Y, Kachouri R, Akil M, Abdallah AB, Bedoui MH. Automated method for real-time AMD screening of fundus images dedicated for mobile devices. Med Biol Eng Comput. 2022 May;60(5):1449-1479. doi: 10.1007/s11517-022-02546-8. Epub 2022 Mar 18. PMID: 35304672.

→ Thank you very much for your kind suggestion. We strongly agree with your comment. From the comprehensive literature review, we have added more details on common indications such as AMD and hypertensive retinopathy. We have also added eight citations to the revised text, including the suggested paper cited in section 4.1.3.

In the text, we have added sentences “For example, Sayadia et al. present an automated method for real-time screening of AMD using fundus images on mobile devices.[38] The proposed system leverages advanced image processing techniques and machine learning algorithms to facilitate rapid and accurate AMD detection, with an accuracy ranging from 94% to 100%. This enables early diagnosis and treatment, potentially reducing the incidence of severe vision loss.[38]” (Lines 298-303) and “4.1.5. Hypertensive retinopathy

The study by Muiesan et al. investigates the feasibility and reliability of using a smartphone-based device (D-Eye) for ocular fundus photography in an emergency department setting for patients with acute hypertension. Results show that the smartphone device significantly detected more abnormal ocular fundus findings com-pared to traditional ophthalmoscopy, suggesting its potential utility in emergency department for better diagnosis of hypertensive emergencies.[41]

4.1.6. Other miscellaneous

Retinoblastoma can be detected using smartphone retinal imaging. As early detection is crucial for effective treatment in patients with this disease, typically young children, smartphone imaging provides a portable and accessible means for early screening.[12]

Ocular Toxoplasmosis, caused by the Toxoplasma gondii parasite, can lead to retinal inflammation and scarring as well as vitritis, leading to significant visual decline. Recurrence is relatively common, which require regular monitoring and surveillance for the patients. Smartphone retinal imaging may be useful to document and monitor the retinal changes associated with this condition, particularly early detection of recurrence.[13]” (Lines 313-328)

Comments on the Quality of English Language

  1. The formatting of cited references should be revised.

→ Our authors have reviewed the journal style of cited references, correcting errors and confirming that the current style and format are in accordance with journal guidelines.

Reviewer 2 Report

Comments and Suggestions for Authors

This manuscript made an introduction of smartphone fundus image. Its advantage and limitations are listed. Different clinical applications are pointed out. However, reviewer believe the comparison lacks quantitative analysis and are too superficial. The work is well organized and well written, can be accepted after making following major revision.

1.      Table 1 is not correctly placed, it’s suggested to be placed in one page for easy reading.

2.      Authors compared advantages and disadvantages for smartphone fundus imaging. Most of them are qualitative analysis, more quantitative analysis will be easier for readers to understand. For example, authors can use pixel amount difference between smartphone and camera to show the variability of image quality.

3.      Some comparisons are too superficial. Like the cost comparison of fundus camera and smartphone. It’s natural to say that smartphone is cheaper with less accuracy, it will be much better if authors can point out why the lower cost and lower accuracy of smartphone is a sweet point of balance making patients and doctors prefer smartphone than camera. For example, decreasing 90% percent equipment cost while only decreasing 10% accuracy may attract patients to use the smartphone methods. And the cost of equipment and cost of use are two things, hospitals may concern the cost of equipment while patients only care how much they need to pay for every use.

4.      For AI in fundus images, there are many factors to consider like the law issues, who should be responsible if the smartphone results are wrong, the limitation of computation resources on smartphone, long inference time, the interest conflict between doctors and smartphone application companies.

Author Response

Reviewer #2

This manuscript made an introduction of smartphone fundus image. Its advantage and limitations are listed. Different clinical applications are pointed out. However, reviewer believe the comparison lacks quantitative analysis and are too superficial. The work is well organized and well written, can be accepted after making following major revision.

  1. Table 1 is not correctly placed, it’s suggested to be placed in one page for easy reading.

→ Thank you very much for your valuable feedback and helpful suggestions. We acknowledge the reviewer's concern regarding the lack of quantitative analysis in our comparisons. We have revised the manuscript to include more quantitative data where available, providing a more robust comparison of the advantages and limitations of smartphone fundus imaging. This includes adding specific metrics and quantitative data from relevant studies to support our discussions. Furthermore, we have reformatted Table 1, which has been shortened as per Reviewer #1’s suggestion, to ensure it fits on a single page for improved readability.

The revised table is as follows:

Table 1. Advantages and limitations of smartphone fundus imaging

Advantages

Limitations

Accessibility and portability

Cost-effectiveness

Familiarity and improved patient engagement

Continuous improvements and updates by continuous software and hardware improvement

Limited image quality, resolution, and field of view

Lack of standardization and calibration leading to variability in image quality and diagnostic accuracy

Regulatory issues for medical devices

Medicolegal issues such as patient privacy, data security, risk of unauthorized access, and potential breaches of confidentiality

This adjustment should make it easier for readers to review the information presented. Thank you again for your constructive feedback.

  1. Authors compared advantages and disadvantages for smartphone fundus imaging. Most of them are qualitative analysis, more quantitative analysis will be easier for readers to understand. For example, authors can use pixel amount difference between smartphone and camera to show the variability of image quality.

→ Thank you for your insightful feedback. We have addressed the concern regarding the need for more quantitative analysis by incorporating specific metrics and quantitative data into our comparisons throughout the manuscript. For instance, we have included data on pixel count differences between smartphone cameras and traditional fundus cameras.

In the text, we have added several paragraphs on the quantitative data as follows:

Conventional digital fundus cameras typically feature image sensors with varying pixel counts, often around 12 megapixels (MP).[24] Modern smartphone cameras may also have high pixel counts; for instance, the study using an iPhone 6 for fundus imaging had an image resolution of 2448 x 3264 pixels (about 8 MP) for single-image acquisition and 1248 x 1664 pixels (about 2 MP) for video mode.[23] However, it is important to note that more pixels don't necessarily translate to better image quality for fundus photography. Due to the optical limitations of the eye itself, there is a point beyond which additional pixels don't improve resolution. The key factors for image quality in fundus photography are often the optics of the system and the eye being imaged, rather than just raw pixel count. Therefore, smartphone-based systems might produce clinically useful images despite potentially lower effective resolutions com-pared to high-end conventional fundus cameras. (Page 4, Lines 139-150)

  1. Some comparisons are too superficial. Like the cost comparison of fundus camera and smartphone. It’s natural to say that smartphone is cheaper with less accuracy, it will be much better if authors can point out why the lower cost and lower accuracy of smartphone is a sweet point of balance making patients and doctors prefer smartphone than camera. For example, decreasing 90% percent equipment cost while only decreasing 10% accuracy may attract patients to use the smartphone methods. And the cost of equipment and cost of use are two things, hospitals may concern the cost of equipment while patients only care how much they need to pay for every use.

→ Thank you for your constructive comments. We recognize the importance of conducting a thorough comparison with conventional camera-based fundus imaging in terms of cost and accuracy to fully grasp the cost-effectiveness of smartphone fundus imaging. For example, we highlight the significant reduction in equipment costs alongside the minor decrease in accuracy, underscoring the practical benefits of smartphone methods. Additionally, we have distinguished between hospital equipment costs and patient per-use expenses to provide a comprehensive understanding of the economic factors influencing doctors' preferences.

In the text, we have added the paragraphs “Cost-effectiveness is another compelling advantage of smartphone fundus imag-ing. Smartphone fundus imaging offers a significant reduction in equipment costs compared to traditional fundus cameras. Traditional fundus cameras, especially those that are table-top models, can be prohibitively expensive, often costing upwards of $10,000 and often require additional costs for maintenance and training.[17] In con-trast, smartphones are widely available and relatively inexpensive, with some setups costing below $1,000.[17] This makes them accessible to a broader range of healthcare providers, particularly in low- and middle-income countries. This dramatic reduction in price makes it feasible to equip more healthcare providers with fundus imaging ca-pabilities, thereby expanding access to essential diagnostic services. The lower cost also opens up possibilities for widespread screening programs, particularly in low-resource settings where the financial burden of traditional equipment has been a barrier.[18]

Moreover, only slight decrease in accuracy observed with smartphone-based methods (e.g., sensitivity and specificity in detecting diabetic retinopathy [DR] and sight-threatening DR) is often within an acceptable range for clinical use. For instance, a study reported a sensitivity of 75.2% and specificity of 95.2% for detecting any DR with a smartphone-based nonmydriatic camera, compared to conventional mydriatic fundus cameras.[19] Another study showed the reliability of an offline AI algorithm for fundus images captured by non-specialist operators using a smartphone-based camera, with a sensitivity of 89.1% and specificity of 94.4% for any DR.[20]

These figures suggest that despite some (usually less than 25%) loss of accuracy, smartphone-based methods still provide robust diagnostic capabilities and thus, de-creasing 90% percent equipment cost for hospitals and the per-use cost for patients while only decreasing 10% accuracy may particularly attract doctors to use the smartphone methods.” (Page 3, Lines 84-108)

Furthermore, with respect to diagnostic accuracy, we have provided quantitative data on smartphone fundus imaging in our text.

The sensitivity and specificity of these devices can vary, but some studies report sensitivity as high as 91% and specificity up to 99% for detecting any DR, making it a viable option for preliminary screening.[6,11,30-32]

More specifically, Ryan et al. evaluated the diagnostic capability of smartphone imaging, finding a sensitivity of 50% and specificity of 94%, which improved to 81% and 94%, respectively, with a non-mydriatic fundus camera.[30] Russo et al. focused on clinically significant macular edema, reporting a sensitivity of 81% and specificity of 98%.[31] Toy et al. examined moderate nonproliferative and worse DR, achieving 91% sensitivity and 99% specificity.[6] Wintergerst et al. reported a sensitivity of 0.79 and specificity of 0.99 for any DR, 1.0 for both measures in severe DR, and 0.79 sensitivity and 1.0 specificity for diabetic maculopathy.[11] Recently, Gobbi et al. found a sensitivity of 71% and specificity of 94% for DR detection.[32] (Page 8, Lines 255-267)

  1. For AI in fundus images, there are many factors to consider like the law issues, who should be responsible if the smartphone results are wrong, the limitation of computation resources on smartphone, long inference time, the interest conflict between doctors and smartphone application companies.

→ Thank you for highlighting these important considerations. We recognize that addressing legal responsibility, computational limitations, inference times, and potential conflicts of interest is crucial for the application of AI in smartphone fundus imaging. We have expanded our discussion to include these factors, providing a more comprehensive overview of the challenges and considerations associated with integrating AI into smartphone-based fundus imaging. We have added the paragraphs

The integration of AI into smartphone-based fundus imaging presents several critical considerations, particularly concerning legal responsibilities, computational limita-tions, and potential conflicts of interest. One significant issue is the legal responsibility when AI-driven diagnostic results are incorrect. Unlike traditional medical examina-tions performed by certified professionals, AI algorithms operate autonomously, rais-ing questions about accountability. If a diagnosis made by an AI on a smartphone leads to incorrect treatment or harm, it is unclear whether liability should fall on the soft-ware developers, the medical institutions utilizing the technology, or the physicians incorporating AI into their practice. This ambiguity necessitates the establishment of clear legal frameworks and guidelines to delineate responsibility and ensure patient safety.[41,51]

Moreover, the computational limitations of smartphones pose another challenge. Although smartphones are ubiquitous and offer a low-cost alternative for fundus im-aging, their limited processing power can result in long inference times for AI algo-rithms. This can be particularly problematic in emergency settings where rapid diag-nosis is crucial. For instance, studies have shown that while smartphone-based devices like the D-Eye can effectively detect ocular abnormalities, the analysis time and image quality can vary significantly.[5,41] Therefore, optimizing AI algorithms for efficiency and ensuring they can operate within the constraints of smartphone hardware are es-sential for practical implementation. Additionally, this requires balancing the need for high-quality diagnostic images with the processing capabilities of the device, which can be a challenging technical hurdle.[3]

Lastly, the introduction of AI in smartphone fundus imaging could potentially lead to conflicts of interest between healthcare providers and technology companies. Physicians might be skeptical of integrating AI tools from commercial entities if they perceive these tools as undermining their professional autonomy or if there are finan-cial incentives involved. Conversely, technology companies may prioritize profit over clinical efficacy, leading to suboptimal patient outcomes. Thus, fostering a collabora-tive environment where both parties work towards common goals, such as improving patient care and ensuring ethical standards, is vital. This collaboration should include transparent communication, shared responsibilities, and continuous education to align the interests of doctors and tech developers.[4,51] In light of this, while the advent of AI in smartphone-based fundus imaging offers promising benefits, it is imperative to address legal, technical, and ethical challenges to ensure its successful and safe imple-mentation in clinical practice.” (Page 13, Lines 465-497)

Round 2

Reviewer 1 Report

Comments and Suggestions for Authors

The authors response all my questions.

Comments on the Quality of English Language

There are some minor grammatical errors, but they do not affect the overall reading

Reviewer 2 Report

Comments and Suggestions for Authors

The Manuscript is well revised. It's interesting to see experts from different fields exchange thoughts on same topics.